

# Impact of Eurasian autumn snow on the winter North Atlantic Oscillation in seasonal forecasts of the 20th century.

Martin Wegmann[1,2,3], Yvan Orsolini[4], Antje Weisheimer[5,6], Bart van den Hurk[6,7] and Gerrit Lohmann[3]

[1]Institute of Geography, University of Bern, Bern, Switzerland.

[2]Oeschger Centre for Climate Change Research, University of Bern, Bern, Switzerland.

[3]Alfred-Wegener-Institute, Helmholtz Center for Polar and Marine Research, Bremerhaven, Germany

[4]NILU–Norwegian Institute for Air Research, Kjeller, Norway

[5]National Centre for Atmospheric Science, Atmospheric, Oceanic and Planetary Physics, University of Oxford, Oxford, United Kingdom

[6]European Centre for Medium-Range Weather Forecasts, Reading, United Kingdom

[7]Deltares, Delft, The Netherlands

Corresponding author: Martin Wegmann, Institute of Geography and Oeschger Centre for Climate Change Research, University of Bern, Hallerstrasse 12, 3012 Bern, Switzerland. Email: martin.wegmann@giub.unibe.ch

**Key points**

Snow–atmosphere coupling, seasonal prediction, North Atlantic Oscillation, polar vortex, stratospheric warming, hindcast





**Abstract**
As the leading climate mode of wintertime climate variability over Europe, the North Atlantic
Oscillation (NAO) has been extensively studied over the last decades. Recently, studies highlighted the
state of the Eurasian cryosphere as a possible predictor for the wintertime NAO. However, missing
correlation between snow cover and wintertime NAO in climate model experiments and strong non-
stationarity of this link in reanalysis data is questioning the causality of this relationship.
Here we use the large ensemble of Atmospheric Seasonal Forecasts of the 20th Century (ASF-20C)
with the European Centre for Medium-Range Weather Forecasts model, focusing on the winter season.
Besides the main 110-year ensemble of 51 members, we investigate a second, perturbed ensemble of
21 members where initial (November) land conditions over the Northern Hemisphere are swapped from
neighboring years. The Eurasian snow / NAO linkage is examined in terms of a longitudinal snow depth
dipole across Eurasia. Subsampling the perturbed forecast ensemble and contrasting members with high
and low initial snow dipole conditions, we found that their composite difference indicates more negative
NAO states in the following winter (DJF) after positive west to east snow cover gradients at the
beginning of November. Surface and atmospheric forecast anomalies through the troposphere and
stratosphere associated with the anomalous positive snow dipole consist of colder early winter surface
temperatures over Eastern Eurasia, an enhanced Ural ridge and increased vertical energy fluxes into the
stratosphere, with a subsequent negative NAO-like signature in the troposphere. We thus confirm the
existence of a causal connection between autumn snow patterns and subsequent winter circulation in
the ASF-20C forecasting system.



## 1. Introduction


As the leading climate variability pattern affecting winter climate over Europe, the North Atlantic
Oscillation (NAO) has been extensively studied over the last decades (Wanner et al., 2001; Hurrell and
Deser, 2010; Moore and Renfrew, 2012; Deser et al., 2017). The NAO state strongly impacts the
hydroclimate as well as the ecological and socioeconomic conditions over major population clusters of
Europe and North America. In its positive state, the NAO projects onto strong pressure gradients over
the North Atlantic, strong westerly winds and mild but wet conditions for Central Europe. A negative
winter NAO is connected to a southwardly displaced Atlantic jet stream, weaker westerlies and cold,
dry conditions for Central Europe. The NAO also shows a distinct quadrupole signature in surface
temperature straddling the Atlantic, with two opposite poles over northern Europe and Greenland
/Labrador and an opposite pair further south over southern Europe/North Africa and the US East Coast.
Recent cases of extreme negative NAO states (Wang and Chen, 2010; Lü et al., 2020), including the
winter 2020/2021, coincided with several extreme weather events across the Northern Hemisphere,
including cold air outbreaks with record snowfall at locations over Southern and Northern Europe, as
well as eastern parts of Canada and the United States.
Improving seasonal to decadal predictions of the winter NAO is a high-priority research for many
weather and climate related research centres (Kang et al., 2014; Scaife et al., 2014, 2016; Smith et al.,
2016; Dunstone et al., 2016; Athanasiadis et al., 2017; Weisheimer et al., 2017; Baker et al., 2018;
Weisheimer et al., 2019). Despite its stochastic behaviour, the NAO state was shown to be modulated
by slowly varying components of the climate system, carrying climate state „memory" across months
or even seasons (Dobrynin et al., 2018; Meehl et al., 2021). Initially discussed by Cohen and Enthekhabi
(1999), recent studies have highlighted the potential of Eurasian autumn snow cover anomalies as a
useful predictor for the boreal wintertime (December– January–February, DJF) NAO in empirical
prediction models (Cohen et al., 2007, 2014; Cohen and Jones 2011; Peings et al., 2013; Tian and Fan
2015; Wang et al., 2017; Han and Sun 2018; Wegmann et al., 2020).
The causal chain behind the snow impact is hypothesized as follows: due to the radiative and
thermodynamical properties of snow (Cohen and Rind 1991; Vavrus 2007; Dutra et al., 2011;
Thackeray et al., 2019), a thicker and more extended snowpack is associated with coherent surface
cooling. Cohen et al., (2007; see also Cohen et al., 2014; Henderson et al., 2018 for reviews) proposed
a multi-step mechanism whereby this surface cooling leads to raised isentropic surfaces, triggering
increased Rossby wave activity propagating upward and being absorbed in the stratosphere, warming
it and subsequently weakening the polar vortex. The negative stratospheric Northern Annular Mode
(NAM) signal eventually propagates down into the troposphere and to the surface where it projects onto
a negative NAO.
Investigating the robustness of this mechanism is challenged by several elements. Observational studies
analyzing statistical links are restricted by the relatively short length (a few decades) of comprehensive



and complete snow cover observations. Using long-term reanalyses, recent studies showed substantial
non-stationary relationships between autumn Eurasian snow cover and the sign of the winter NAO over
the span of the 20th century (Peings et al., 2013; Douville et al., 2016; Wegmann et al., 2020). Using
shorter time scales, the probability of "cherry picking" a period of increased correlation and sampling
co-variability with other climate system components increases considerably. Causes for the non-
stationarity are still discussed, with possible influences from the Quasi-Biennial Oscillation (QBO), El
Niño-Southern Oscillation (ENSO) or simply snow cover variance (Peings et al., 2017; O'Reilly et al.,
2017; Tyrrell et al., 2018; Wegmann et al., 2020; Weisheimer et al., 2020). Disentangling co-variability
is further challenged by the co-occurrence of increased Eurasian snow cover and increased Ural
blocking frequency, questioning the lead-lag relationship between snow cover and blocking (Peings
2019; Song and Wu, 2019; Santolaria-Otín et al., 2021). Moreover, a variety of temporal and spatial
snow cover indices used among the different studies obstruct direct comparisons. Nevertheless, recent
studies point out that a November longitudinal snow cover dipole across Eurasia shows the strongest
statistical link to the DJF NAO state (Gastineau et al., 2017; Han and Sun 2018; Santolaria-Otín et al.,

95   2021).

Analyzing the snow → NAO mechanism in modelling experiments is challenged by short-comings of
the current Atmospheric or Atmosphere-Ocean General Circulation Models (AGCMs or AOGCMs)
regarding snow-atmosphere feedbacks (Santolaria-Otín and Zolina 2020). Most of the free-running
Coupled-Model Intercomparison Project (CMIP) models do not capture the statistical snow-NAO link
found in reanalyses data (Hardimann et al., 2008; Furtado et al., 2015; Gastineau et al., 2017). On the
other hand, when large snowpack anomalies are prescribed through nudging or imposed as initial
conditions, several AGCM experiments showed promising results for identifying several to all steps of
the proposed mechanism (Gong et al., 2003; Fletcher et al., 2009; Peings et al., 2012; Tyrrell et al.,

104   2018).

Some of the current-generation subseasonal-to-seasonal or seasonal coupled prediction models also
seem to catch parts of the mechanism chain, specifically negative temperature anomalies associated
with a thicker snowpack (Orsolini et al., 2013; Diro and Lin, 2020) as well as an enhanced wave activity
generating upward fluxes into the stratosphere associated with ridging over western or eastern Eurasia
(Orsolini et al., 2016; Li et al., 2019; Garfinkel et al., 2020), although several models failed to simulate
that ridging in the latter multi-model study. The subsequent stratosphere-troposphere coupling
influencing the surface Arctic Oscillation also tended to be weak to non-existent in most models. These
studies have been limited to the recent decades and, consequently, confidence in the robustness of the
mechanism across spans of decades is still low and needs to be strengthened (Garfinkel et al., 2020).
To disentangle the issues of non-stationarity (found in observations) and causality (found in models),
we base our investigation on a 110-year long (1901-2010) ensemble seasonal prediction experiment,
which is based on the historical seasonal forecast initiative using ECMWF's atmosphere-only model,



called "ASF-20C" (Weisheimer et al., 2017). This 51-member ensemble experiment with four start
dates per year and a forecast length of 4 months has been used in several studies on the predictability
of the NAO and other climate patterns (e.g., O'Reilly et al., 2017; Parker et al., 2019; Weisheimer et
al., 2019; Weisheimer et al., 2020; O'Reilly et al., 2020). To investigate the influence of land surface
conditions, in this case snow cover, on the evolution of the atmospheric state throughout the season, we
use a novel, 21-member twin set of the ASF-20C forecasts with perturbed initial land conditions. This
dataset was used as a pilot experiment in the context of the Land Surface, Snow and Soil moisture
Model Intercomparison Program LS3MIP (Van den Hurk et al., 2016), aimed at reproducing land
surface potential predictability experiments as described by Dirmeyer et al. (2013). We aim to address
the question of causality, pathway, stationarity and seasonal evolution of the proposed mechanism of
the snow-stratosphere-troposphere linkage over decadal to centennial time scales.
This paper is organized as follows. Sect. 2 describes the data and methods used. In Sect. 3, we show
winter evolution of climate anomalies for the different initialization runs and contrast them with
observed anomalies. The results are discussed in Sect. 4 and finally summarized in Sect. 5.





## 2. Data and Methods

### a. Climate reanalysis and reconstruction

We use the Centre for Medium-Range Weather Forecasts (ECMWF) product ERA-20C (ERA20C; Poli et al., 2016) to investigate pre-conditions and the initialization of the seasonal predictions, to compute the DJF NAO index as well as to create a Eurasian snow dipole index. ERA-20C only assimilates surface pressure and marine wind observations, with sea surface temperature (SSTs) boundary conditions taken from the HadISST2.1.0.0 datasets (Rayner et al., 2003). ERA-20C was found to represent interannual snow variations over Eurasia remarkably well. For an in-depth discussion of its performance and the technical details concerning snow computation, see Wegmann et al., (2017). Due to the a-forementioned statistical impact for the winter NAO evolution, we focus on the November Eurasian snow dipole index as predictor for the following NAO state (Gastineau et al., 2017; Han and Sun 2018; Santolaria-Otín et al., 2021). Following Han and Sun (2018), we calculate the index over the period 1901–2010 by averaging snow depths over the western domain (30°-60°E, 48°N-58°N) and the eastern domain (80°-130°E, 40°-56°N), eventually subtracting the western domain from the eastern domain to derive the west-east snow cover gradient. Hence, a positive snow index indicates higher snow depths in the eastern domain and a positive longitudinal snow gradient. The index is normalized and linearly detrended with respect to the overall time period. To comply with the initialization date of 1$^{st}$ of November for the seasonal prediction runs, we calculate the index for 1$^{st}$ of November instead of November mean snow (Figure 1). Even though Han and Sun (2018) calculated the dipole index using snow cover, we used snow depth since ERA-20C provides snow depth as the actual prognostic variable. We hence refrained from using empirical rules to convert snow depth to snow cover. We found the index based on snow depth to be virtually identically (also see Supplementary Figure S1) to the index using snow cover (see also Wegmann et al., 2020 for more insights).

To compute the winter NAO index, we normalize the first Empirical Orthogonal Function of ERA-20C DJF sea level pressure (SLP) for the region (90°-50°E, 20°-80°N). We use the same definition for the NAO DJF index in seasonal prediction runs and compare those with the reconstructed, independent DJF NAO index by Jones et al., (1997) from the Climate Research Unit (CRU).

### b. Seasonal prediction experiments

Additionally, we use atmospheric seasonal retrospective predictions covering the 110-year period 1901-2010 of ERA-20C with 51 ensemble members of the ASF-20C hindcasts (hereafter ASF-20C CTL) (Weisheimer et al., 2017). The atmospheric model used for the 4-month forecasts is the ECMWF Integrated Forecast System Version CY41R1 and is initialized at four start dates per year (1$^{st}$ of Feb, May, Aug and Nov) with ERA20C land and atmospheric conditions. It uses the same lower boundary conditions for SST and sea ice as ERA-20C. Here, we only use forecasts initialized on the 1$^{st}$ of November. The horizontal spectral resolution of the model of T255 is similar to ECMWF's previous operational system System 4 (Molteni et al., 2011) and corresponds to a grid length of approximately



80 km. The model has 91 vertical levels and a top at 0.01 hPa. The ensemble has been created by
perturbing each member through the stochastic physics schemes to represent model uncertainties in a
similar way as the a-forementioned System 4.
To investigate the impact of Eurasian snow depth we use an additional set of perturbed forecasts, based
on a 21–member subset of the ASF-20C CTL experiment (hereafter, the "Experiment" or ASF-20C
EXP). Each member run is initialized with different land surface conditions, sampled from the
neighbouring 20 years. For example, the range of land surface conditions for the 21-member ensemble
forecast initialised on $1^{st}$ of November 1950 spans the land surface conditions of the years 1940–1960:
member 01 is initialized with the land surface conditions of 1940, member 02 of 1941, member 03 of
1942 and so forth. For the beginning and ending ten years of the hindcast dataset, the land surface
conditions are sampled from the closest 21 neighbouring years within the dataset. Here, land surface
conditions include the entire land state, including soil moisture, snow depth and soil temperatures. We
argue that for investigating northern hemisphere climate anomalies of the $1^{st}$ of November initialization,
snow depth has by far the largest impact on atmospheric dynamics compared to soil moisture and soil
temperatures, thus allowing us to attribute the differences to snow changes. The main bulk of the
experiment data has a monthly resolution, daily data is only available for selected variables and three
tropospheric levels.
Taking advantage of the shuffled initial land conditions of ensemble members in ASF-20C EXP, we
subsample members with positive or negative initial Eurasian snow dipole (Figure 2). This conditional
sampling approach has been used when testing the sensitivity of extended range forecasts to soil
moisture (Koster et al., 2011; van den Hurk et al., 2012) or to snow initial conditions (Li et al., 2019;
Garfinkel et al., 2020). For each start date, we can identify those members with positive or negative
initial snow dipole indices, corresponding to different years of the shuffled land initialisation. We
further proceed with compositing these two selected sets. Due to the decadal variability in the November
snow cover, the amount of "high snow members" (positive dipole index) and "low snow members"
(negative dipole index) varies throughout the 110 years. There might be periods when a majority of the
neighbouring 20 years shows a positive snow dipole index and other periods when a minority does. To
avoid this variation of the composited ensemble size across the years, we only use the five ensemble
members with the most positive and most negative initial Eurasian snow dipole, creating two ensemble
means (each of size N=5), namely a high snow dipole ensemble-mean and low snow dipole ensemble
mean, for each winter through the 110-year period.
It should be noted that the absolute magnitude of the ensemble-mean snow differences is still changing
from year to year. For example, the most positive snow dipole for the period 1910–1930 might be lower
than in the time window 1980–2000, and the same applies for negative dipole indices. Due to the
definition of the ASF-20C EXP, this setup is unavoidable, but it also allows for realistic magnitudes of



snow forcings and for incorporating a realistic natural variability into the experiment. The (5-member)
ensemble-mean difference (Figure 3a) displays a snow depth increase of 1-2 cm over Central and
Eastern Siberia, together with a 0.2-1 cm snow depth decrease over Western Russia, as expected from
the snow dipole definition. Concomitant negative anomalies (1-2 cm snow depth) nevertheless extend
outside of the dipole definition domains to more northern latitudes, e.g., over Western Russia and the
Russian Far East, or over the coastal mountain ranges of the North American Pacific Northwest. Note
that these two domains are in snow transition zones, where the snow cover is rare on November 1$^{st}$ and
shows some variability (Figure 3b-c). The location of the sub-sampled snow forcing however adds snow
towards Eastern Eurasia, in locations where the ERA20C snow depth climatology computes a few
centimeters of snow. In contrast, snow removal to the west of Russia appears in regions with no to rare
snow cover in the ERA20C November 1$^{st}$ climatology.  The eastern domain partly covers the Mongolian
Plateau region which was shown to exert a strong impact of the wintertime wave fluxes in the
stratosphere [White et al., 2017].
If not stated otherwise we compute differences between the 5-member ensemble means of the "high
snow dipole" and "low snow dipole" in ASF-20C-EXP as well as differences of each ensemble mean
relative to the ensemble mean of ASF-20C CTL. We compute significance using a two-sided Student's
t-test.

## 3. Results

### a. DJF NAO comparison

Figure 4a shows the time series of the normalized reconstructed (i.e., based on station data), reanalysed
and predicted winter NAO state for the period 1901–2010. Unsurprisingly, the ensemble means of the
ASF-20C CTL and ASF-20C EXP forecasts show reduced temporal variance compared to the
observation-based NAO datasets. However, single realizations and member spread of the CTL and EXP
runs cover the whole range of variability displayed by the observation-based product.
The correlation between the ERA-20C and CRU NAO index is 0.83, indicating that the EOF approach
is a good approximation of the station-based index. It should be noted that the DJF average has a higher
correlation between forecasts and reanalyses than the individual monthly correlations within the season
(see Supplementary Table S1).
The ASF-20C CTL ensemble mean DJF forecast achieves an overall correlation of 0.33 with the CRU
NAO reconstruction for the complete time period, with ASF-20C EXP having a nearly identical
correlation (0.34). This near-identical correlation is expected given that the land state perturbations
across the 21 members are two-sided. Differences between the predicted NAO index of ASF-20C CTL
and EXP ensemble means are generally small, with the NAO indices having the same sign during most





winters. The correlation between CTL and EXP is 0.8 for the 110-year period. The slightly stronger
variability of ASF-20C EXP can partly be attributed to the reduced ensemble size.
Contrasting the (initial) high-snow and low-snow composites constructed from the ASF-20C EXP
ensemble, we see decadal variability in the difference of winter-mean NAO (Figure 4b&c). The first
two decades of the 20[th]-century are characterized by rather strong negative NAO responses to a strong
positive snow dipole. This is followed by two decades spanning the early twentieth century Arctic
warming, which shows the opposite response to the multi-step mechanism hypothesized above: A
positive snow dipole, as depicted in Figure 4, leads to more positive NAO states compared to a negative
snow dipole forcing. After several decades with changing responses to the snow dipole between the two
ensembles, eventually the 21[st]-century starts with a weak negative NAO response to a strong positive
snow dipole. Averaged over the whole period, the high-snow ensemble shows a slightly stronger
negative NAO response, which is pronounced for extreme NAO states with 51(18) cases of positive
(+1 SD) NAO response, 59 (29) cases of negative (-1 SD) NAO response. For 2 SD exceedance, the
number of cases is 2 vs 9. Possible reasons for the decadal response to the snow forcing will be
considered in the discussion section.

**b.   Regression analysis**

Previous studies showed that regressing observed boreal winter zonal-mean temperature and zonal wind
anomalies onto an observed Eurasian autumn snow index reveals a significant stratospheric warming
and slow-down of the polar vortex starting in November, migrating down towards the tropopause until
February. A similar relation between Eurasian snow and the polar stratosphere can be found in the
dataset used here.
Figure 5 shows a strongly reduced polar vortex for the ERA20C autumn to winter climate anomalies
regressed on the November snow dipole index. The zonal wind anomalies in the troposphere highlight
a weakened polar jet and an increased subtropical jet, especially in January and February. The
concurrent polar stratospheric warming signal moves towards the upper troposphere throughout the
winter months, with peak warming at around 100 hPa in February.
Spatially, pressure anomalies regressed onto the November snow dipole index reveal that the
geographical center of the stratospheric warming is located over the Canadian Arctic (Figure 6).
Tropospheric pressure differences highlight a strong ridging over Western Russia and the Ural
Mountains in December, which subsequently over the course of winter is shifted more towards
Greenland and the Northern North Atlantic region, reflecting a negative NAO-like atmospheric state.
This state is further supported by negative DJF SLP anomalies over Southern Europe and the
Mediterranean region. Downstream of the Eurasian snow signal, a negative SLP anomaly is found over
the Northern North Pacific. The question remains, if these patterns are a result of sampling random co-
variability or if the statistical analysis is indeed capturing physical processes.



### c. Spatial anomalies in the experiment

In the following paragraphs we investigate the spatial differences in the atmospheric response associated with the high-snow and low-snow ensemble means of ASF-20C EXP, focusing on the initial response in December as well as the average DJF response.

Figure 7a&b shows stratospheric geopotential heights anomalies at 10 hPa. In December, a significant negative anomaly formed above Eurasia, corresponding to a polar vortex displacement toward the Eurasian sector and a high over Alaska, as commonly found during stratospheric warming events. Over the course of the winter, this pattern develops into increased geopotential heights over the Arctic with significantly reduced geopotential heights over the extratropics.

To better understand the wave activity flux into the stratosphere, we investigated the meridional eddy heat flux at 100 hPa, which is proportional to the vertical component of the wave activity flux (Figure 7c): it highlights a wave train of circumpolar anomalies in December (hence, following the surface signal forcing in November) with significant positive anomalies over the Ural mountains, eastern North Pacific and the European part of the North Atlantic and negative anomalies over Central and Northern Europe and along the North American Pacific coast. The average DJF response highlights a circumpolar wave-train but shows significant anomalies only for the increased northward heat flux over the northern North Atlantic.

Tropospheric circulation anomalies are depicted for geopotential heights at 500hPa in Figure 7e&f. In December, a strong positive anomaly is located over the Barents-Kara Sea sector, with significantly negative anomalies up- and downstream. A second region of positive anomalies emerges at the Canadian Atlantic coast. Both regions match the significant positive anomalies in the 100 hPa heat flux well. The averaged DJF anomalies highlight a negative mid-tropospheric NAO signal with significantly increased geopotential heights above Greenland and Iceland.

Sea level pressure anomalies largely mirrors the 500 hPa geopotential height anomalies. The averaged DJF pattern only shows significant increased anomalies over the northern North Atlantic, but still projects onto a meridional pressure gradient characteristic of a negative NAO phase (Fig. 7h). It is important to note, that the absolute difference is rather small compared to interannual SLP variability. Anomalies between the two ensemble-means are around 1 hPa. Even though this number can be assumed to be smaller than in observational datasets due to the ensemble averaging process, it only constitutes ca. 15% of the average 1901–2010 DJF SLP standard deviation over the Euro-Atlantic sector.

Due to its large variability, composites of the near-surface temperature are largely non-significant (Figure 7i&j). Yet, in December a clear cooling signal emerges over Central and Eastern Eurasia, as expected from the location of the positive snow anomalies at the time of forecast initialization. At the



same time, eastern North America and south-eastern Europe show significant positive temperature
anomalies, a result of northward heat advection at the eastern flanks of low-pressure anomalies (Figure
7g). Averaged DJF 2m temperatures are significant only for Greenland and Eastern Eurasia, with the
cooling over the latter a direct result of the persistence of the anomalously high initial snowpack.
**d.  Vertical anomalies in the experiment**
To get a better understanding on how the different initial conditions impact the vertical distribution of
temperature and zonal wind, Figure 8 shows meridional cross-section of the zonal-mean anomalies of
zonal wind and temperatures from November to February.
While November anomalies (Figure 8) are overall insignificant, a strong snow dipole is associated with
an increased polar vortex and cooler stratosphere. In December, zonal wind anomalies are indicative of
the tropospheric subtropical jet shifted northward concurrent with a weak Arctic surface warming.
Changes are substantial in January, when the stratospheric polar vortex is significantly weakened, with
a slight increase in westerlies in the mid-troposphere. The corresponding temperature anomalies show
a widespread stratospheric warming and negative anomalies in the lower Arctic troposphere. Eventually
in February, the slow-down of westerlies is predicted to reach all the way down from the stratosphere
into the troposphere. On the southern flank of these negative zonal wind anomalies, westerly winds are
increasing, especially so in the stratosphere. The stratospheric warming signal migrates downwards to
the lower stratosphere and tropopause layer. As the warming has migrated down, a stratospheric cooling
is forecasted aloft.
As a further confirmation, polar cap heights (Supplementary Figure S2) reveal a development of
positive anomalies from the surface in December up to the stratosphere in January, migrating back to
the troposphere in February.
**Daily evolution of anomalies in the experiment (?)**
To investigate the temporal evolution and importance of tropospheric anomalies, Figure 9 shows daily
mean meridional mean 500 hPa GPH anomalies (high minus low snow dipole ensembles) averaged
over 60-70°N. The Hovmöller diagram illustrates the Ural ridge developing only at the end of
November going into December and is pre-ceding the development of the North Atlantic ridge, which
is the main component of the negative NAO-like feature in our results. It should also be noted that the
absence of meaningful anomalies during the first ten days of the composite difference reflects the
subtraction of tropospheric anomalies arising from the pre-conditions (since atmospheric initial states
are identical among the perturbed ensemble members). The anomalies generated by the end of
November do indeed arise from the impact of snow cover differences and snow-atmosphere feedbacks.
**e.  Non-linearities in the snow forcing impact**



Two distinct non-linearities need to be considered. First, a non-linearity in the physical snow feedback:
adding a few centimetres of snow in a snow–covered region will not change the radiative and
thermodynamic properties of the already snow-covered land surface substantially (due to a saturation
effect) but, by contrast, removing a few centimetres of snow might remove the snow layer altogether,
changing the albedo and thermodynamics of the surface–atmosphere boundary. This non-linearity may
be important for the Rossby wave generation as air flows over the uplifted isentropes above the snow-
covered area. The non-linear effect of snow cover saturation and the impact of the relative magnitude
of regional surface cooling in our experiments is addressed by Figure 10. In years when the high-minus
low snow cover anomalies resulted in a negative NAO anomaly (see Figure 4c for indication of years),
the December cooling anomaly over Eastern Eurasia is much stronger than for the opposite case when
they resulted in a positive NAO anomaly. Concurrently, the formation of a Ural ridge anomaly is much
more pronounced, flanked by troughs up and downstream, with positive eddy heat fluxes into the
stratosphere over the Barents-Kara Sea and widespread stratospheric warming. This supports the notion
that adding an absolute amount of snow (in either of the two longitudinal domains) is not sufficient for
the causal chain to be triggered. Instead, it is a large (in magnitude and extent) relative surface impact
of the additional snow that triggers the initial anomalous Rossby wave generation part of the
hypothesized causal chain.

A second non-linearity is the asymmetrical role of the eastern and western domains of the snow dipole.
Our subsampling of the ASF-20C EXP simulation allows to estimate the respective roles of these two
domains. Interestingly, the difference between the low-snow ensemble mean and the CTL ensemble
mean for DJF sea level pressure (Figure 11) reveals a much stronger response to a negative snow dipole
(i.e., with high snow depths over Western Russia and low snow depths over Eastern Eurasia) than to
the positive snow dipole (i.e., with high snow depths over Eastern Eurasia and low snow depths over
Western Russia). In other words, an anomalously negative NAO signal in boreal winter in the high-
snow minus low-snow anomalies is mostly a result of an anomalously positive NAO signal in the low-
snow ensemble. The negative dipole corresponds to lower snow depths over the eastern domain
(Mongolian Plateau and surroundings areas), consistent with lessened wave fluxes into the stratosphere
over this region which is the important orographic driver of climatological upward wave fluxes in winter
(White et al., 2017). On this note, additional snow right around the Ural Mountains (a negative dipole
index) does not enhance the pre-existing role of the Ural mountains in Rossby wave generation by much
in our setup.
A possible reason for this non-linear behaviour might be found in the importance of the Rossby wave
generation via the Eastern Russia cold air dome over the snow-covered area. Based on the snow depth
climatology of Eurasia (Figure 3), the negative dipole forcing represents a much more severe disruption
to the climatological snow distribution than the positive index forcing is. Removing snow over Eastern



Eurasia seems to favour zonal flow more than adding snow is favouring meridional flow. Again, the
relative impact of the snow forcing is key in this context. Interestingly, additional snow upstream or
just right around the Ural mountains (a negative dipole index) does not seem to enhance the pre-existing
role of the Ural mountains in Rossby wave generation by much and we do not find positive the
geopotential height anomalies in its vicinity.

### 4. Discussion

We used a set of centennial ensemble seasonal forecasts (ASF-20C) and a complementary set with
perturbed land initial conditions (ASF-20C-EXP) to address some of the open questions regarding the
relationship between Eurasian autumn snow cover and the state of the NAO in the following winter.
Subsampling of the latter forecast set according to the initial value (on $1^{st}$ of November) of a west-east
snow dipole over Eurasia (Gastineau et al., 2017; Han and Sun, 2018) allowed us to determine the
response over 110 winters.
The regression of stratospheric wind and temperature upon the snow dipole in ERA20C over the 1901-
2010 period reveals a weakened stratospheric vortex in January and February, following a positive
initial snow dipole. The seasonal evolution of the ASF-20C EXP high- minus low-snow anomalies
similarly indicates a weakened polar vortex. It also supports the notion of a surface cooling over the
Eastern domain anchoring a Ural ridge anomaly on its western flank in December (Figure 7e). This
Ural ridge triggers an increased northward heat flux in the lower stratosphere, thereby reducing the
polar vortex strength and increasing polar stratospheric temperatures. In January and February, the
signal moves downwards into the troposphere where it evolves into a negative NAO anomaly. In
general, these results agree with the framework proposed by Cohen et al., (2007) and the experiments
with the ECMWF seasonal prediction model by Orsolini et al. (2016). However, it should be highlighted
that the absolute ensemble-mean, time-average SLP signal, diagnosed as the conditional composite
difference in ASF-20C EXP, is very small, about 1 hPa. As mentioned before, this represents only a
small fraction of the interannual SLP variability in the Euro-Atlantic region. Nevertheless, for single
realisations of winter forecasts, this impact can be much higher.
The role of the Ural ridge in the snow cover → NAO causal chain has been discussed and analysed in
several recent studies (Peings 2019; Santolaria-Otín et al., 2020). Here we find that the Ural ridge is a
pre-condition of predicted negative NAO winters in ASF-20 CTL (Supplementary Figure S3), together
with a cold 2m temperature anomaly in Eastern Russia and a cold stratospheric polar vortex displaced
over Eurasia, downstream of the Ural ridge. However, these initial conditions are subtracted out in the
ASF-20C EXP high- minus low-snow composite difference, and we find that the composite difference
indicates a re-enforced Ural ridge (Figure7e). We find the mid-troposphere Ural ridge is re-inforced
only at the end of November going into December, which pre-cedes the formation of a North Atlantic
ridge that prevails until February (Figure 9). This result indicates that the snowpack does indeed play a
feedback role (see also Orsolini et al., 2016). Thus, we propose that the relation between the Ural ridge





and Eurasian snow cover consists of a mutual interaction: the circulation anomaly associated to a pre-
existing Ural ridge shovels cold polar air southwards along its eastern flank, allowing for an extensive
snow cover to form over Eastern Eurasia (eastern domain of the dipole). In addition to this process
(Figure 10c), our analysis reveals that the snow cover anomaly re-enforces the Ural ridge, allowing for
increased wave flux into the stratosphere. This particular location of a tropospheric ridge interferes
constructively with climatological stationary wave-1 and wave-2 patterns (Garfinkel et al., 2010) and
seems to be key for a skilled forecast of the polar winter stratosphere (Portal et al., 2021).
Furthermore, the high minus low composite highlights a subpolar North Pacific surface  and mi-
tropospheric low-pressure anomaly that appears first in December and remain dominant throughout all
of DJF (Figures 7e,f,g,i and 11). The generation of this circulation feature was pointed out by previous
studies (Orsolini and Kvamstø, 2009; Garfinkel et al., 2010; Garfinkel et al., 2020), and has been
attributed to an enhanced vertical propagation of Rossby waves into the stratosphere and horizontal
downstream of the cooled Eurasian land mass.
Subsampling of the experimental multi-decadal historical forecasts (ASF-20C EXP) highlighted an
interdecadal variability and non-stationarity of the snow dipole impact, despite the cancelling out of
common boundary forcings such as SST evolution in the composite difference. The configuration of
our experiment does not allow to explain this behaviour completely; however, we can address some
possible reasons. A potential influence on the decadal variability of the snow cover impact might be the
precursory climate system state, promoting or counteracting the tendency for the (perturbed) snow
forcing towards a given NAO state.
Surprisingly, the positive snow dipole forcing tends to favour a negative NAO signal when the climate
system is "tuned" for a positive winter NAO, for example when high Barents-Kara sea ice extent and
La Niña SST conditions prevail (Supplementary Figure S4). This supports the idea of a clear and strong
snow cover impact when the regional cooling anomaly in Eastern Eurasia is relatively strong and the
climate state is preconditioned to a rather positive NAO-like condition. Such snow conditions -
corresponding to a positive snow dipole - might amplify the snow feedback in the atmospheric
prediction system, tugging the NAO towards a more negative state (while the absolute state might still
be positive). This might explain the strong positive anomaly during the early twentieth century Arctic
warming in Figure 4c: the period 1920–1940 was characterized by a strong positive mid-tropospheric
high anomaly from Northern Europe to East Siberia (Wegmann et al., 2016). We find that the 500 hPa
anomalies between high and low snow composites show only a weak to non-existing Ural ridge for the
period 1921–1940, when compared to e.g. 1991–2010 (Supplementary Figure S5). On the opposite,
adding (removing) snow in Eastern (Western) Eurasia to provide the same response as the pre-existing
anomalies favoured by other background conditions, does not seem to have a linear impact. Rather,
strong non-linearities seem to occur, which is reasonable given the non-linear thermodynamic and
radiative impacts of increased snow cover.



On that note, we find that the relative magnitude of regional cooling compared to the existing climate
state in our experiments is of crucial importance. In years when the high-minus low snow cover
anomalies resulted in a negative NAO, the December cooling anomaly over Eastern Eurasia is much
stronger than for the opposite case (Fig. 10). Moreover, we found that a negative snow dipole forcing
leading to a positive NAO signal has a much larger relative impact compared to the positive snow dipole
resulting in a negative NAO signal. In our experimental setup, snow removal in the eastern domain and
snow increase in the western domain allows for a rather zonal circulation in the following months, with
no subsequent stratospheric warming signal. Distinct model experiments are needed to understand the
atmospheric feedbacks of these configurations better, however it should be kept in mind for future
studies using regression or similar statistical tools to infer about the impact of Eurasian snow cover.
As such, we find that the main driver for the proposed snow-stratosphere linkage is a large relative
impact of the additional snow cover, especially on surface temperatures. Generally, our results further
highlight the importance behind the land memory effect discussed by Nakamura et al. (2019), who
argue for a delayed impact of snow cover via soil and surface temperatures.
Nevertheless, we are limited in analysing the impact of co-variability in the climate system over the
span of the 110-year period. Additional experiments are needed to investigate the role of climate state
precursors and memory effects influencing the seasonal predictions.
## 5. Summary and Conclusion
Centennial seasonal ensemble forecasts were used to examine the impact of an increased November
Eurasian west-to-east snow cover dipole on the boreal winter climate evolution. We found evidence for
the manifestation of a negative NAO signal after a positive November west-to-east snow cover dipole
via surface cooling, increased Ural blocking and subsequent stratospheric warming (although evolution
toward a positive NAO state was also observed but less frequently, especially for NAO extremes).
Including 110-years of natural Earth System variability increases the confidence in the proposed
physical mechanisms behind cryospheric drivers of atmospheric variability and decreases the
probability of random co-variability between snow cover and DJF NAO. Our results hence support
previous hypotheses and statistical studies. The absolute surface impact was found to be small in our
experimental setup, with interdecadal variability and ensemble averaging reducing the magnitude of
individual events.
We found the impact of our snow forcing to be strongest for climate states that will allow the snow
forcing to exert a strong surface cooling. Adding additional snow on top of already existing snow-
covered and cold surfaces does not linearly strengthen the negative NAO development.
Future studies need to address the interplay between different Earth System components in coupled
seasonal prediction experiments. How important the background conditions of the climate system are
before the initialization of the forecasts needs to be investigated further. Furthermore, allowing more



extreme snow forcing (e.g., perturbing initial land states over a longer range than the neighbouring 10
years) might result in stronger stratospheric and surface signals.

**Data availability:**
The ERA-20C reanalysis data is publicly available (https://apps.ecmwf.int/datasets/). The NAO
reconstruction is publicly available at the Climate Research Unit repository
(https://crudata.uea.ac.uk/cru/data/nao/). The ASF-20C dataset is publicly available at the CEDA
Archive (https://catalogue.ceda.ac.uk/uuid/6e1c3df49f644a0f812818080bed5e45). The ASF-20C
experiment dataset (created by Bart van den Hurk) can be made available on the ECMWF MARS
system upon request.

**Author contribution:**
MW and YO designed the study. MW analysed the data. AW and BvH provided the data. GL provided
discussion and interpretation of the results. All authors contributed by interpreting the results and
writing the paper.

**Acknowledgements:**
The authors thank Daniel Balting for coding advice and Judah Cohen for fruitful discussions. GL and
MW recieved funding through BMBF for the topic "Ocean and Cryosphere under climate change" in
the Program "Changing Earth - Sustaining our Future" of the Helmholtz society.

**Competing Interests:**
The authors declare no competing interests.

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



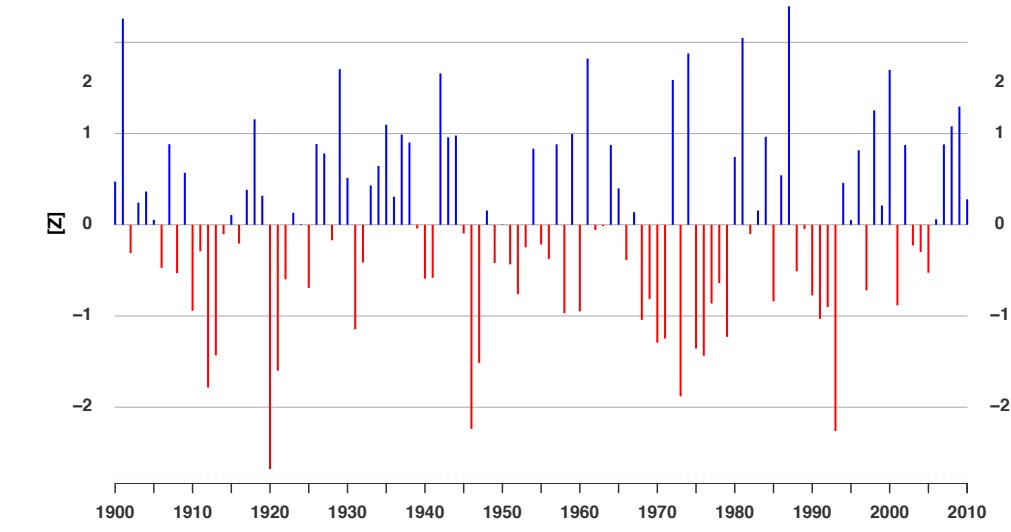

*Figure 1: Normalized 1st of November Eurasian snow dipole index for the period 1900–2010 as derived from ERA20C.*



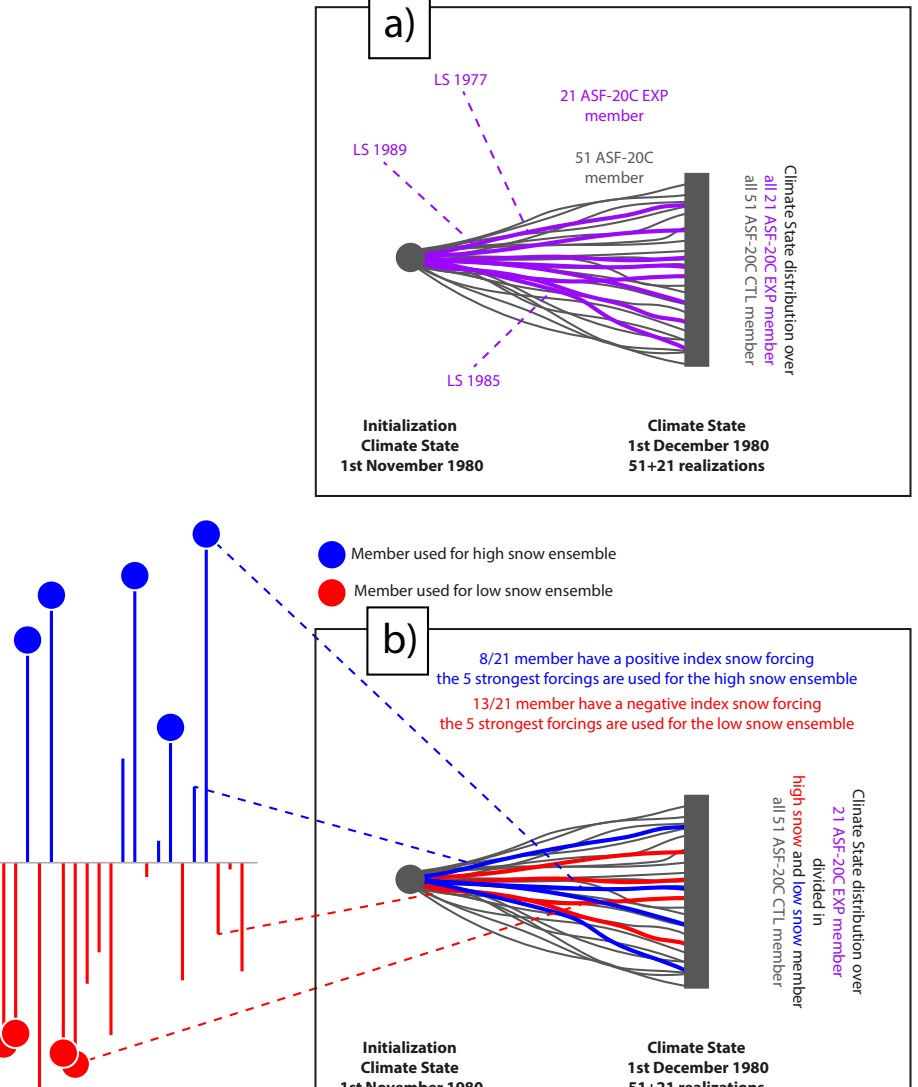

Figure 2: As example, the schematic for a) the 1980 1st of November ASF-20C EXP initialization and
the consequent sampling of the 21 ensemble members into the high and low snow dipole ensembles. For
the 1st of November initialization, ASF-20C EXP members are initialized by land surface conditions of
the 21 surrounding 1st of November dates, in this case 1970–1990, b) Out of these 21 members, we
sample individual members based on their ranking in the snow index. The five members with the most
positive snow index constitute the high snow ensemble and vice versa for the low snow ensemble.



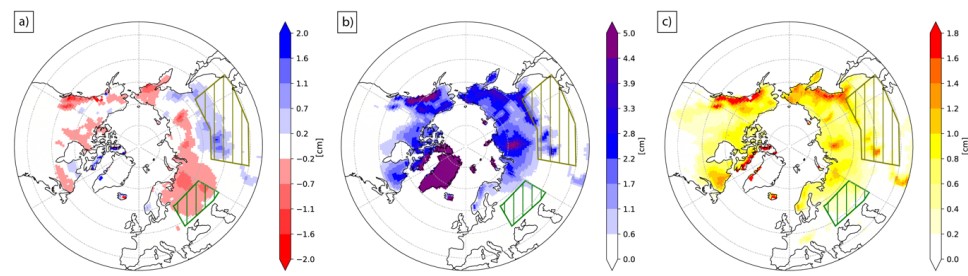

*Figure 3: a) Average (1900–2010) 1ˢᵗ of November snow depth difference between the high-snow and*
*low-snow ensemble. b) Average (1900–2010) 1ˢᵗ of November snow depth. c) Average (1900–2010) 1ˢᵗ*
*of November snow depth standard deviation. Hatched in green (olive) is the western (eastern) domain*
*of the snow index. All 3 plots are based on ERA20C.*



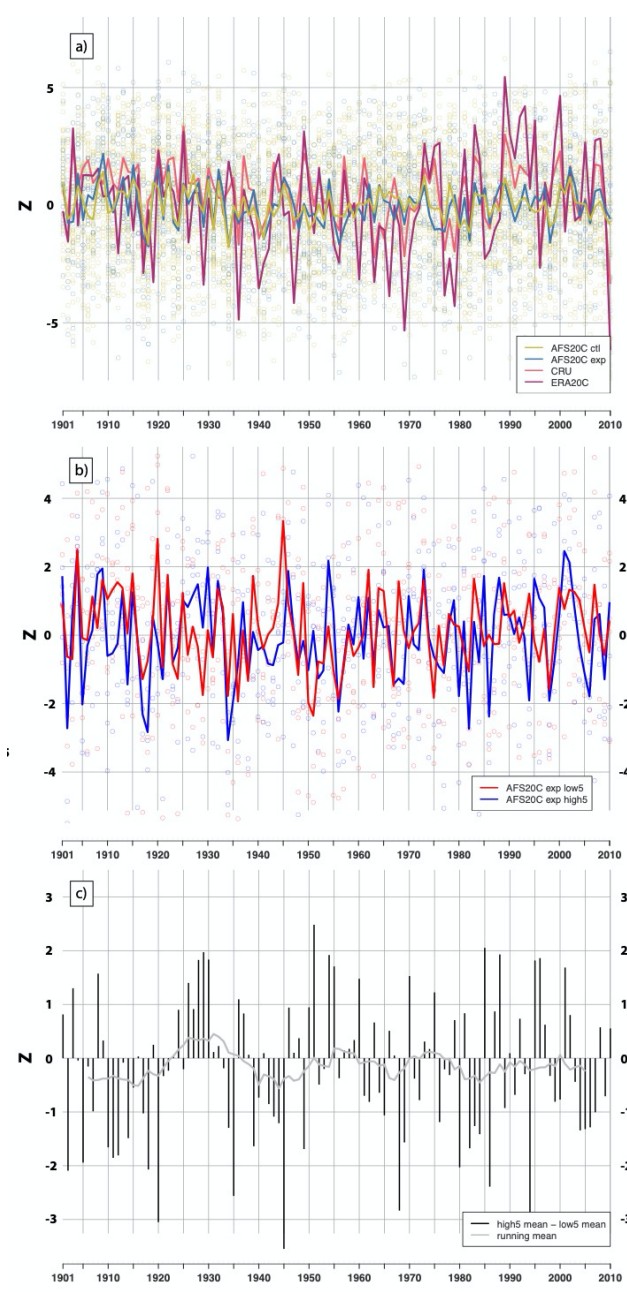


Figure 4: Time series of DJF NAO for the period 1901–2010. a) Normalized DJF NAO index in the
CRU station-based reconstruction, ERA20C EOF-based index, ASF-20C CTL and ASF-20C EXP EOF-
based index. Hollow points represent individual member, solid lines represent ensemble means or
observational products. b) 5-member DJF NAO forecasts for the high- and low-snow members within
ASF-20C EXP. Hollow points represent individual member, solid lines represent ensemble means. c)
NAO DJF state difference and its 11-year running mean between the ASF-20C EXP high- and low-



*snow ensemble mean in panel b (51(18) cases of positive (+1 SD) NAO response, 59 (29) cases of*
*negative (-1 SD) NAO response). For 2 SD exceedance, the number of cases is 2 vs 9.*

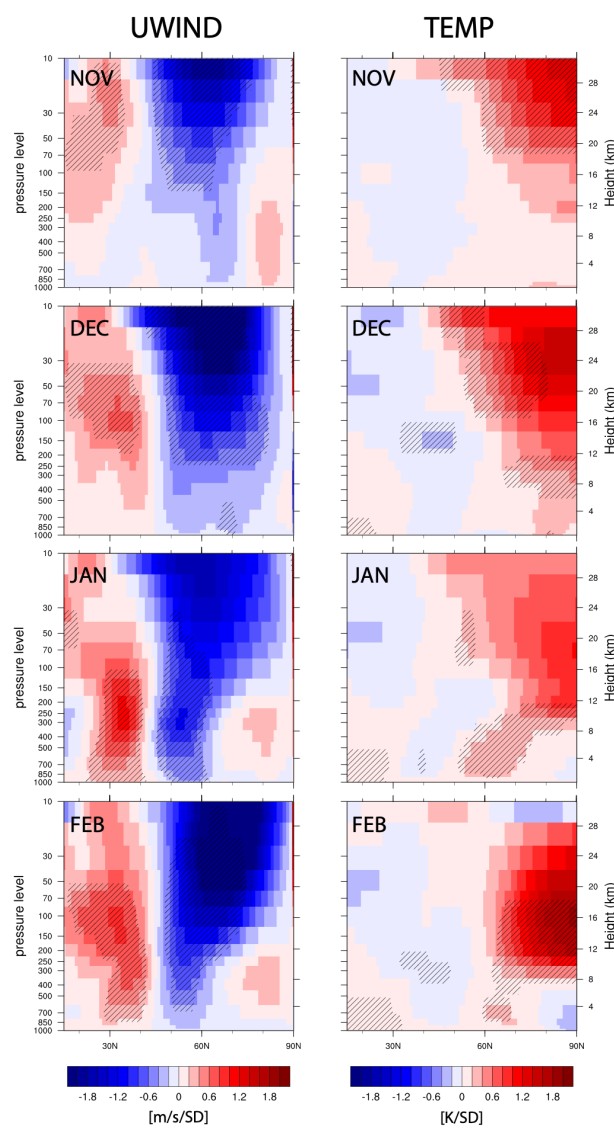


*Figure 5: Zonal-mean meridional cross-section of ERA20C anomalies in temperature and zonal wind*
*regressed onto the snow dipole index in November from ERA20C covering 1901–2010 for November,*
*December, January and February. Shading indicates 95% significance level.*

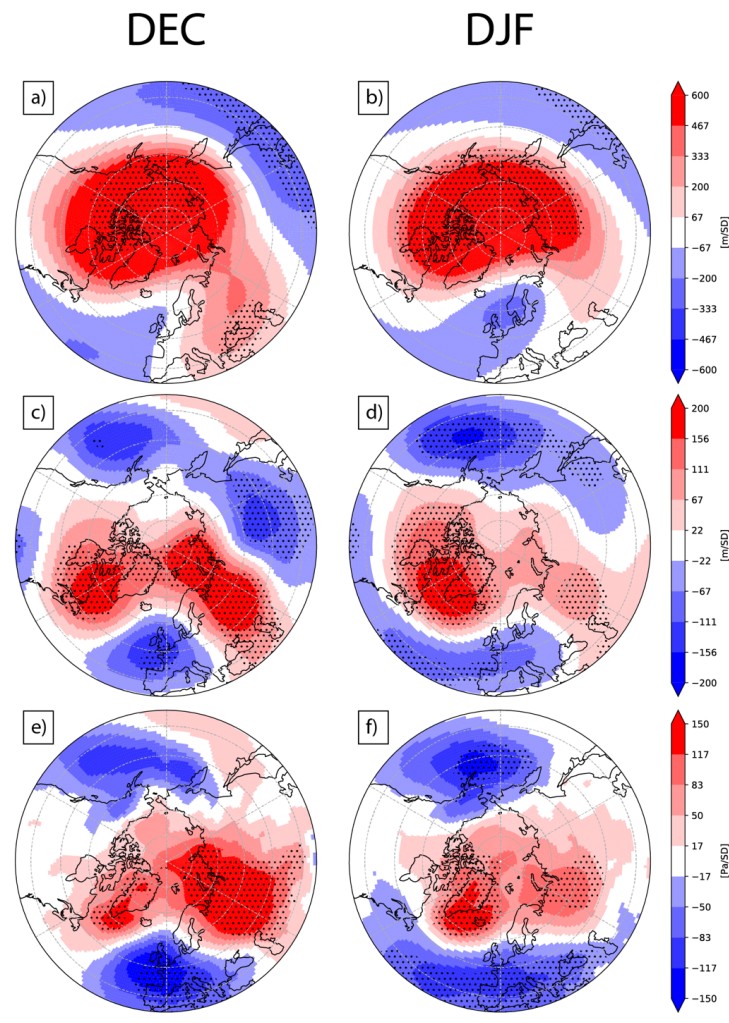


*Figure 6: ERA20C anomalies of a&b)10 hPa geopotential heighst, c&d) 500 hPa geopotential heights and e&f) Sea Level Pressure regressed onto the snow dipole index in November from ERA20C covering 1901–2010 for December and DJF mean. Shading indicates 95% significance level.*

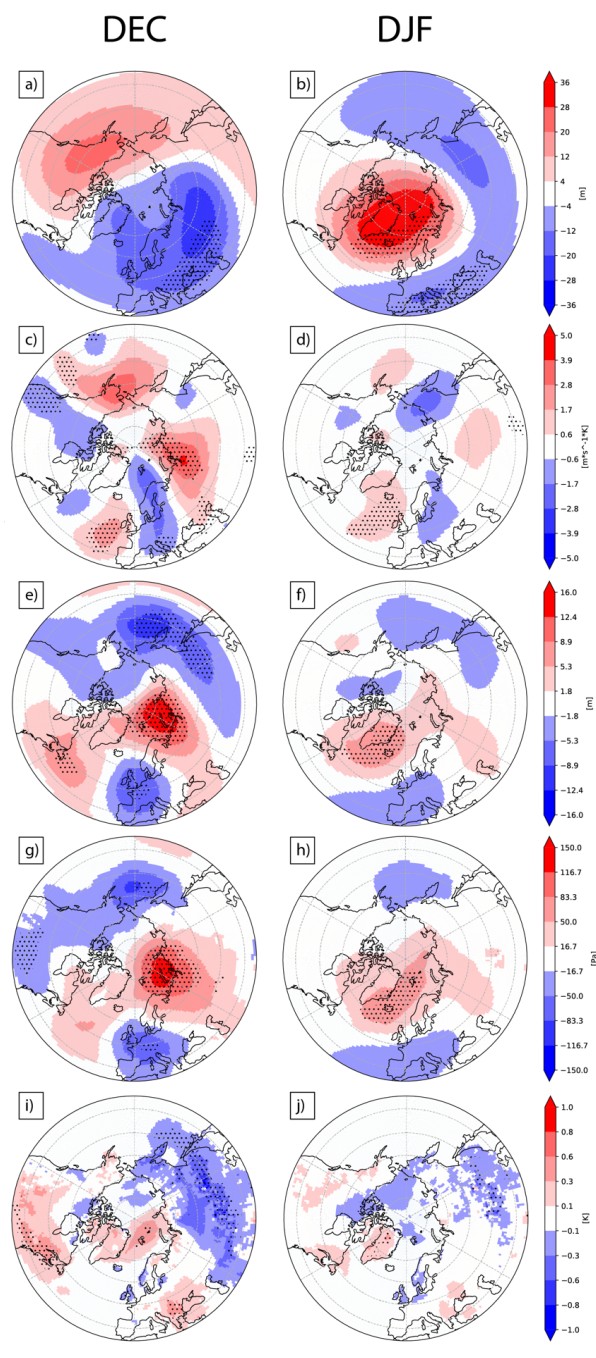


Figure 7: Averaged anomalies 1901-2010 between high-snow and low-snow ASF-20C EXP ensemble
means for December (a,c,e,g,j), and DJF (b,d,f,i,k): a&b) 10 hPa geopotential heights, c&d) 100 hPa
meridional eddy heat flux, e&f) 500 hPa geopotential heights, g&h) sea level pressure and i&j) 2m
temperature. Stippled areas represent 90% significance.

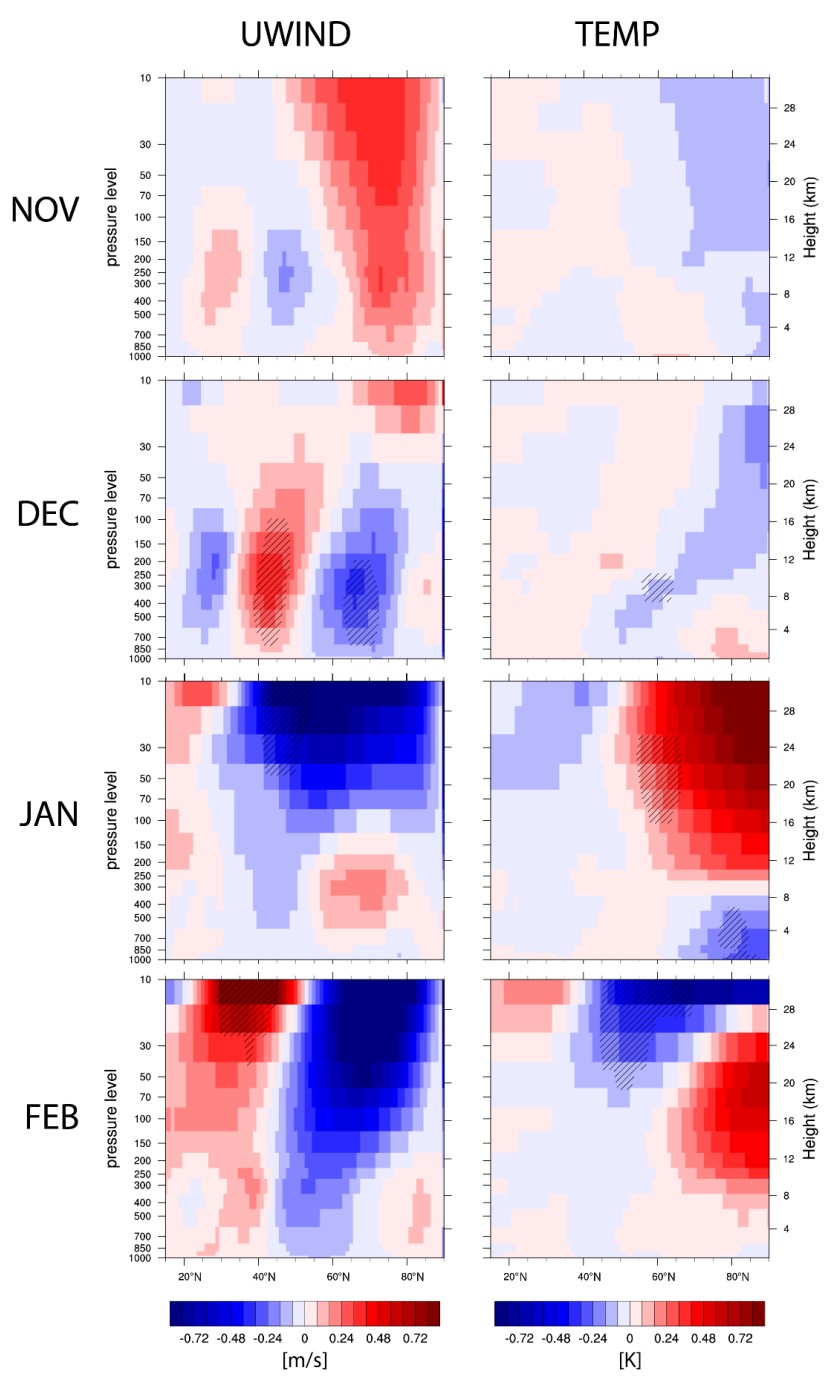


*Figure 8: Zonal-mean cross-section of left) zonal wind anomalies and right) temperature anomalies for
the period 1901-2010 between high-snow and low-snow ASF-20C EXP ensemble means. Shading
indicates 90% significance level.*

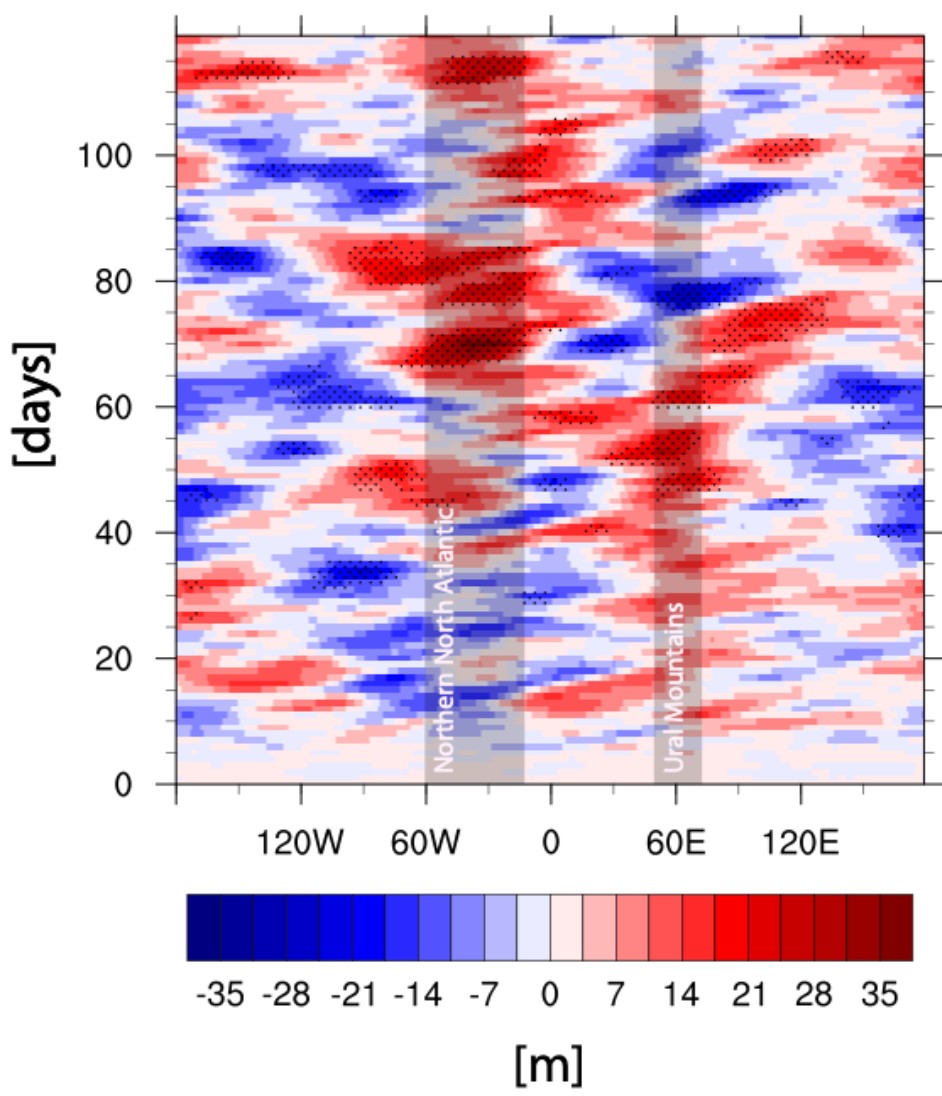


*Figure 9: Hovmøller diagram of daily mean predicted 500 hPa  geopotential height anomalies for the*
*period 1901-2010 averaged for the latitude band 60°-70°N difference between high-snow and low-snow*
*ASF-20C EXP ensemble means. Stippled areas represent 90% significance. Days from NOV 1$^{st}$ are*
*indicated on y-axis.*



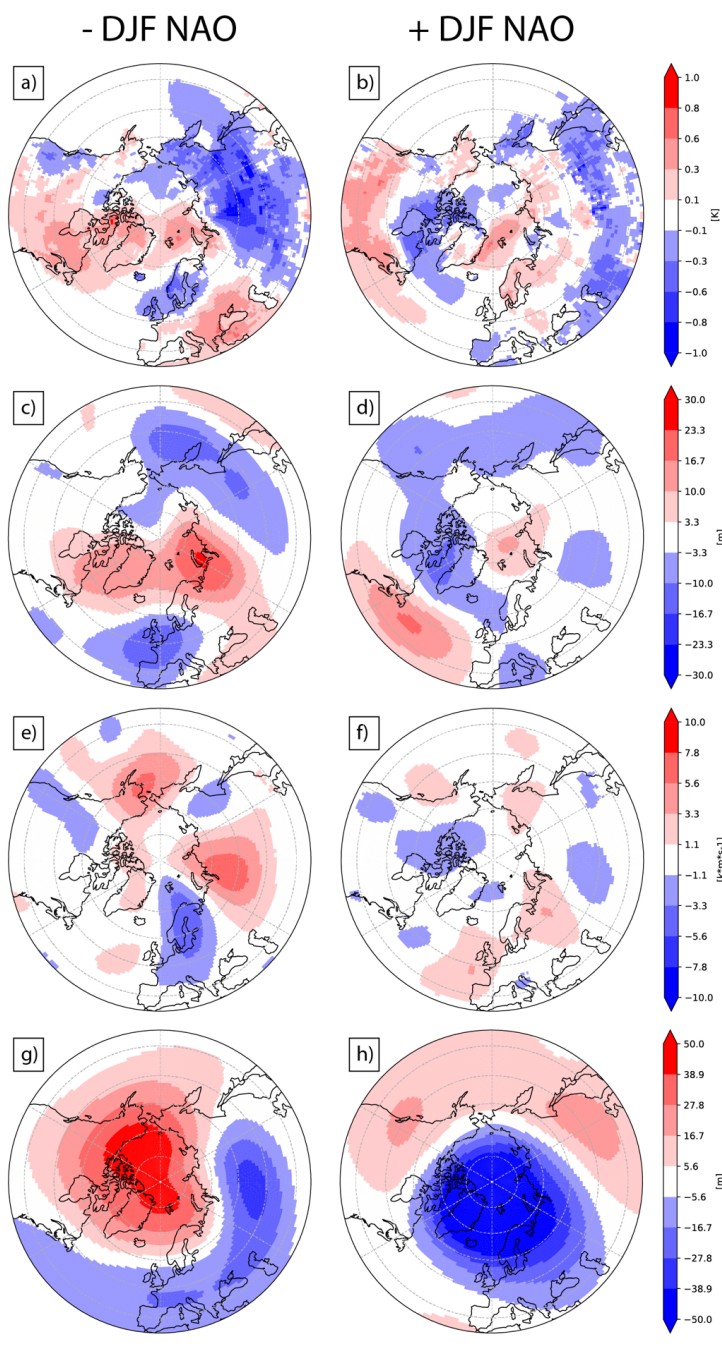


Figure 10: Climate anomaly composites of predicted December fields after which a positive snow dipole
forcing resulted in a negative DJF NAO signal (a,c,e,g) or a positive DJF NAO signal (b,d,f,i)(selection
of years based on Figure 4c): a&b) 2m temperature, c&d) 500 hPa geopotential heights, e&f) 100hPa
meridional eddy temperature flux, g&h) 10 hPa geopotential heights. Anomalies are based on ASF-
20C EXP high minus low snow ensemble mean data.

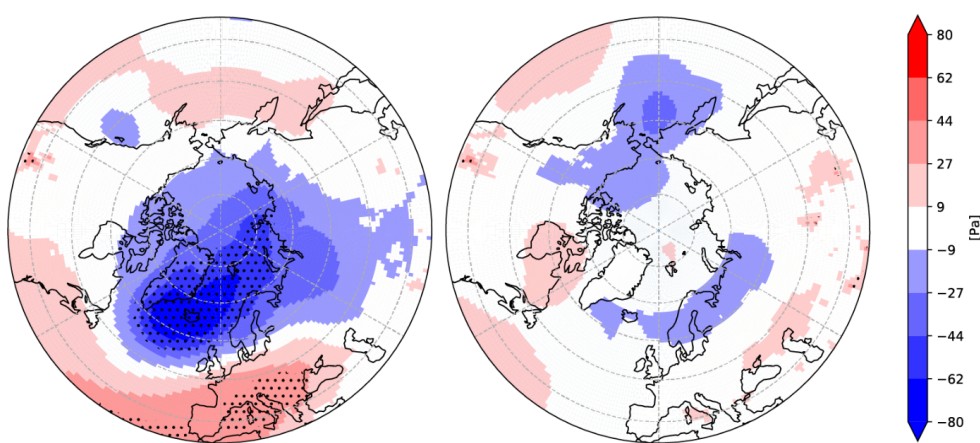

763

*Figure 11: Mean sea level pressure [unit] DJF anomalies for the period 1901-2010 between a) low-snow ASF-20C EXP ensemble mean and ASF-20C CTL ensemble mean (subsampled from 21 CTL members) and b) high-snow ASF-20C EXP ensemble mean and ASF-20C CTL ensemble mean (subsampled from 21 CTL members). Stippled areas represent 90% significance.*

768