# Peer review of "Impact of Eurasian autumn snow on the winter North Atlantic"

_Weather and Climate Dynamics, 2021_

## Author Response (AR1)

REV#1

This paper uses a large ensemble of seasonal forecasts to address open questions about the potential link between Autumnal Eurasian snow cover and the winter North Atlantic Oscillation (NAO).

I find this to be a convincing, carefully thought out, and well written paper, with important conclusions. I would recommend publication subject to addressing the two minor comments below.

MINOR COMMENTS

1)

If the mechanism of Autumn Eurasian snow influencing the winter NAO were not real, I'm not sure I'd suggest that the signal seen in reanalysis was "random co-variability between snow cover and DJF NAO", as is suggested in this paper a few times. Much more likely, I'd have thought, is that the same external driver is influencing both the Autumnal Eurasian snow cover and the winter NAO.

For example Arctic sea ice can influence Eurasian snow cover

https://agupubs.onlinelibrary.wiley.com/doi/full/10.1029/2019JD030339

and also the NAO

https://www.nature.com/articles/s41598-017-00353-y

Tropical rainfall has also been shown to strongly influence the extratropics on seasonal time scales

https://rmets.onlinelibrary.wiley.com/doi/full/10.1002/qj.2910

Can you demonstrate that both sea ice and tropical rainfall are near-enough identical in your high snow and low snow composites? If you can then that would be a good result to report too. It would arguably rule out the above suggestion of either sea ice or tropical rainfall influencing both snow and the NAO, and would strengthen your conclusion that it is snow influencing the NAO.

If, however, you find that sea ice or tropical rainfall are different in your high and low snow composites (due, presumably, to different land surface conditions outside of Eurasia), I think you need to demonstrate somehow that neither sea ice nor tropical rainfall plays an important role here.

REPLY: Thank you very much for your comment. In our experiment, sea ice and SSTs are not dynamical. As such, sea ice is the same for each of the EXP members and is subtracted out as forcing. We checked the tropical precipitation response in the EXP members and found no significant anomalies for November and only sparse significant precipitation spots in December. As such we rule out the impact of tropical rainfall on the winter NAO in our experiments. We changed the wording of "random co-variability" throughout the paper and added the information about sea ice and tropical precipitation to the discussion section. Line

2)

A caveat of previous modelling experiments that prescribe snowpack anomalies, is that the snow anomalies required to produce a significant NAO signal are unrealistically huge. I think you do much better with your experiments here, and indeed think that this is a key strength of the paper (perhaps you could emphasize this point). Therefore, I'd just like it clarified that the anomlies in your high and low snow composites are indeed realistic.

You state that "snow removal to the west of Russia appears in regions with no to rare snow cover". Does that leave you with negative snow anomalies in your low snow composites, which would be unrealistic, or do you set the snow to have a minimum value of zero? I think this needs to be clarified. [Admittedly, it shouldn't impact your conclusions since you show in section 3e that it is the negative dipole (with additional snow to the west of Russia) that leads to most of the response seen.]

Also in the last sentence of the paper you talk about "allowing more extreme snow forcing" but I think you should clarify that this forcing needs to remain realistic (otherwise any conclusions would not add to previous modelling experiments).

REPLY: Thank you very much for your comment. We adjusted the "unrealistic" point throughout the document. Snow depth can never go below 0 in our composites and experiments. Our wording was not precise enough. We subsample members with low snow depth in the west and compare them with members with high snow in the west. We made sure that this wording is improved throughout the document.

In previous experiments with the ECMWF seasonal forecast model (Orsolini et al., 2013), the snow anomalies were not unrealistic but rather taken from different times in the seasonal cycle (e.g, October $1^{st}$ forecast could have been initialized using December $1^{st}$ snow for some members). Here, the snow intialisation is more consistent as it is the same date for all members (but corresponding to different years)

VERY MINOR COMMENTS AND TYPOGRAPHICAL ERRORS

Line 110: "latter multi-model study". It is not clear to me which study you are referring to here.

REPLY: Fixed

Line 134: "We use the Centre" should be "We use the European Centre"

REPLY: Fixed

Line 142: "as predictor" should be "as a predictor"

REPLY: Fixed

Line 153: "identically" should be "identical"

REPLY: Fixed

Line 332: Delete "(?)"

REPLY: Fixed

Line 383: "positive the" should be "positive"

REPLY: Fixed

Figure 2 caption: "As example" should be "As an example"

REPLY: Fixed

Figure 4 caption: "individual member" should be "individual members"

REPLY: Fixed

REV#2

Summmary

This study uses a novel experimental setup that allows the authors to test the impact of different Eurasion snow states on the seasonal evolution of the polar vortex and NAO. Regression analysis and compositing high and low snow dipole scenarios are used to show a relationship between 1st November Eurasion snow and the NAO.

Overall I think this is a well written paper, and an interesting study with a novel use of the exerimental setup. My only concern is that a discussion of non-linearities in the results seems to imply a large difference between the impact of high and low snow, and this difference may be hidden by the use of linear regressions. I believe this should be explained, or investigated briefly, before publishing.

Major Points

Section 4 e. Non-linearities, Fig. 11. This was very interesting to separate the effects of a high and low snow dipole, but then it seems to call into question the core result of the paper, i.e. the statement from the abstract: "Subsampling the perturbed forecast ensemble and contrasting members with high and low initial snow dipole conditions, we found that their composite difference indicates more negative NAO states in the following winter (DJF) after positive west to east snow cover gradients at the beginning of November." this statment is still true, but it seems to imply that high snow leads to a negative NAO (i.e. the canonical view of the snow-polar-vortex-NAO connection). But then Fig. 11b doesn't show a signal for high snow. Does this imply that the regressions, e.g. in Fig. 6 would also include this strong non-linearity (which, of course, would not be picked up the a linear regression). Perhaps this could be explored with a scatter plot, or something similar (I trust the specific details to the authors discetion), showing the snow dipole index plotted against indices of the NAO or an appropriate polar vortex index. If these plots showed a fairly linear response of NAO/polar vortex to a high/low snow dipole, then I think the conclusions are valid. If there is a strong non-linearity and the NAO response is mostly driven by low-snow, then perhaps the conclusions need to be modified somewhat, but this it still a very interesting result.

REPLY: Thank you very much for your comment. We modified Figure 11 (now Figure 10) now and added two additional supplementary figures plus changed the wording about this feature substantially throughout the document. That said, here are possible explanations.

As you rightly said, our anomalies are still true in the model world, just as we described them. Now we should have been more precise with Figure 11, which we are now. In Figure 10a we basically compare a highly modified snow depth distribution with "reality" and in Figure 10b we compare a "realistic" snow depth distribution with reality. It follows naturally, that the stronger change of the earth system gives us a stronger signal. New information in Figure 10 now goes more in depth and shows that the high snow state is indeed able to generate increased sea level pressure over the Atlantic albeit with much lesser impact than the low snow state does.

This makes sense in the physical world. Adding a bit of snow on a snow free surface has a much higher impact than removing a bit of snow from a thick snowpack. Physically, the impact of added snow on the western domain has to be bigger than added snow on the eastern domain. All in all, we would argue for 1) a strong longitudinal gradient has a chance of modifying the NAO towards a more negative state and 2) a weak longitudinal gradient always prefers a positive NAO, which is the preferred state by the Earth System.

Now that is the model world. In the regression we try to model this gradient by defining an index. What we see is indeed a similar behaviour. Weak gradients favours positive NAOs. We also find that a linear regression model using only the eastern domain snow depth variability in explaining DJF NAO shows less significance than a model only using the western domain snow depth variability. The west-east gradient shows the highest significance for predicting wintertime NAO, no matter if we use ERA20C derived NAO or station-based NAO. For a new and more in depth discussion of this topic see Chapter 3f.

Minor points

Would it be practical to show Figure 1 and 4b together? Since it is interesting to compare these.

REPLY: We combine Figure 1 and Figure 4 in the new manuscript.

Line 64: "memory" (quotation marks in english are always up top)

REPLY: Fixed

Line 94 and 141: Is there a reason, or proposed mechanism, for why a snow dipole is a better index than a snow extent index used in many earlier studies, and can this be briefly mentioned in the introduction?

REPLY: Thank you very much for your comment. Just below Line 94 we refer to previous studies that showed that the November snow gradient or dipole is the strongest snow co-variate with DJF NAO, rather than a uniform large scale snow depth field or an October snow depth metric. Studies cited are Gastineau et al., 2017; Han and Sun 2018; Santolaria-Otín et al., 2021. Based on their findings, we decided that it makes most sense to use this November snow depth longitudinal gradient for our experiment setup.

Line 171: For land surface perturbed runs I assume the SSTs, sea ice, etc, are not changed, can this be stated.

REPLY: Thank you very much for your comment. Indeed, SSTs and sea ice are not changed. We added that information to the text, see also Line 425–430.

Line 244: "positive snow dipole, as depicted in Figure 4," Do you mean Figure 3?

REPLY: Fixed

Line 248. Do the numbers in the brackets mean the values for plus/minus one standard deviation? Please write as a separate sentence if it's important.

REPLY: Fixed

Figure 6. Rather than Dec and DJF, perhaps showing Dec, Jan, Feb would be better so we can see the seasonal progression of Z and slp anomalies. If Nov was included too then we couls also see the SLP associated with the snow anomalies.

REPLY: Thank you very much for your comment. Our first initial concept included to show the individual months but we decided to focus on the most important information. Since the two other reviewers seem to be okay with that concept, we will stay with it for the time being. However, we added the monthly information for Figure 6 (new 5) and Figure 7 (new 6) now in the Supplement, so that the interested reader has access to the more detailed information. We do not show the November response, since as shown in Figure 9 (new 8), significant anomalies really only emerge at the beginning of December. This is true for almost all variables.

Line 276: Why is the initial response shown for December, and not November?

REPLY: We do not show the November response, since as shown in Figure 9, significant anomalies really only emerge at the beginning of December. This is true for almost all variables. We added that information to the text.

Figure 6, 7, 10, Supp Fig 4: Short titles above. or on the left side of each plot with the variable, e.g. 500hPa Z, SLP etc, makes it easier to read, compared to relying on the captions.

REPLY: Thank you very much for your comment. We added the variable information to the plots.

Line 303: "ca." I think this abbreviation is usually used for historical dates. It's not incorrect, but perhaps just using "about" would aid coprehension for most audiences.

REPLY: Fixed

Line 319, Fig 8: Is the tropospheric jet shift related to the snow anomaly?

REPLY: All the anomalies in Figure 8 are based on the AFS-20C EXP subsampling and compositing and, as such, result from the snow gradient difference.

Line 336/413: "preceding", "precedes"

REPLY: Fixed

Figure 11: missing 'a' and 'b'

REPLY: Fixed

Line 448: Using parentheses to describe opposite effects can make sentences very difficult to understand, see article: Robock, A. (2010), Parentheses are (are not) for references and clarification (saving Space), Eos Trans. AGU, 91(45), 419–419, doi:10.1029/2010EO450004. https://eos.org/opinions/parentheses-are-are-not-for-references-and-clarification-saving-space

REPLY: Fixed

REV#3

This paper examines relationships between autumn Eurasian snow conditions and subsequent winter NAO development and associated climatic conditions over a 110-year span in the ERA-20C reanalysis, the ECMWF ASF-20C seasonal hindcasts, and a tailored hindcast set in which land surface initial conditions, including snow, are sampled from 20 adjacent years. It is found that differences in the longitudinal gradient of Eurasian snow at the beginning of November have discernable influences on the subsequent winter NAO, but that this relationship is not stationary over the 110-year period and is weaker in the hindcasts than in the reanalysis. Anomalies composited on extreme values of the longitudinal snow gradient point to roles played by the Ural ridge and wave fluxes influencing the stratosphere.

Overall, the paper makes interesting contributions to efforts to unravel the hypothesized causal connection between autumn Eurasian snow distribution and winter climate in the Northern Hemisphere. Although the methodologies and conclusions drawn appear generally robust, the paper could benefit from improvements to the presentation and some additional discussion of certain points as recommended below.

Main comments:

1) In the first paragraph of the introduction at line 57, regarding the NAO and its impacts in the winter of 2020/21 the authors could cite https://doi.org/ 10.1175/2021BAMSStateoftheClimate.1 which touches on this on p. S73 and in Fig. 2.40.

REPLY: Added the reference

2) The following cited papers (may not be a complete list) are missing from the references: Meehl et al., 2021; Diro and Lin, 2020; Wegmann et al., 2017; Wegmann et al., 2016; Jones et al., 1997; Koster et al. 2011.

REPLY: Fixed

3) Near line 143, please say a few words about why these particular eastern and western domains were chosen so that the reader does not have to refer to Wegmann et al., 2017 (which is not in the reference list as commented above). Presumably their latitudinal range is intended to encompass variations in the November 1 snow line? (This is explained finally around line 209, so maybe could add "whose choices are motivated below" or something similar at line 145.) Also, is it possible to represent the domains in Fig. 3 accurately, with borders along latitude circles rather than inaccurately as line segments?

REPLY: Thank you so much for your comment. In the introduction we refer to previous studies that showed that the November snow gradient or dipole is the strongest snow covariate with DJF NAO, rather than a uniform large scale snow depth field or an October snow depth metric. Studies cited are Gastineau et al., 2017; Han and Sun 2018; Santolaria-Otín et al., 2021. Based on their findings, we decided that it makes most sense to use this November snow depth gradient for our experiment setup. However, we added your suggestion to the Methods section to highlight how our motivation is linked to previous studies. We fixed Figure 3.

4) Inconsistent terminology is used to describe the index for Eurasian snow distribution. For example, in section 2a the eastern minus western difference in Eurasian snow depths is described as the "west-east snow cover gradient". However, snow cover as conventionally defined refers to the presence or absence of snow (irrespective of depth), and snow cover extent to area blanketed by snow. To avoid potential confusion, and for consistency with the discussion on lines 150-154, I suggest referring here to the "west-east snow depth gradient" or simply "west-east snow gradient".

REPLY: We fixed the inconsistent wording to "west-east snow depth gradient" throughout the manuscript.

5) Although the derived index is consistently and appropriately called the Eurasian snow dipole index or simply dipole index, the sub-ensembles of high and low dipole index hindcasts are referred to the "high-snow ensemble" and "low-snow ensemble". However, it's not obvious what "high snow" and "low snow" refer to, and calling these the "high-dipole ensemble" and "low-dipole ensemble" would be clearer and more consistent.

REPLY: We fixed the inconsistent wording to "high-dipole" and "low-dipole" throughout the manuscript.

6) At lines 156-157, it's not entirely clear whether using the "same definition for the NAO DJF index in seasonal prediction runs" means that the predicted NAO index is obtained using the first EOF of ERA-20C SLP, or the first EOF of predicted SLP.

REPLY: Clarified this point. We use the first EOF of predicted SLP.

7) It should be clarified in section 2b that these are not, strictly speaking, seasonal retrospective predictions since the ocean and sea ice boundary conditions after initialization are based on observations (same as ERA-20C) rather than being predicted using damped anomaly persistence or some other means.

REPLY: Thanks for pointing this out, we added this information.

8) In my opinion Fig. 2 would be easier to process if the three "LSxx" labels in panel (a) and associated dashed lines were removed, since their meaning isn't explicitly defined and it's not obvious exactly what the dashed lines are connecting to. (Although their meaning can be distilled from the main text, I found these features to be more distracting than illuminating, e.g. one wonders why there are three and not some other number, why those particular years, etc. all of which are irrelevant to the point of the figure.) More broadly, it might be commented that it may take most readers longer to understand the figure than the descriptions of the two experiments in section 2b which are straightforward, although the figure does nicely schematize what was done.

REPLY: Thanks for pointing this out. Since the two other reviewers didn't mention a possible removal of Figure 2, we keep it in the manuscript but removed the dashed lines and added a bit of explanation in the new version.

9) Considering that the western domain typically has little snow on November 1 (lines 212-213), is the dipole index time series, and by extension the overall results of this study, much different if only variability in the eastern domain is considered? (The discussion of Fig. 11 on page 12 also suggests this might be the case.)

REPLY: Thank you very much for your comment. We do not have the capacities to repeat the whole analysis with a different subsampling procedure, but there are certain hints we checked and can check. 1) Fig 11 (now Fig 10) rather suggests that the western domain carries most of the information. That does make sense, since the eastern domain will most likely remain snow covered, no matter the snow dipole index, whereas the western domain can easily change from no snow cover to fully snow covered from year to year. 2) We can check your question in the statistical sense in ERA20C and the real world. We checked a linear regression between DJF NAO and a) the eastern domain, b) the western domain and c) the difference (or gradient) and found that the eastern domain has consistently the lowest significance for predicting DJF NAO, no matter if we use ERA20C derived NAO or station-based NAO. The western domain carries more significance and finally the gradient showing the best model out of these three options. As such, I would answer your question with: Yes, it would look much different. We added additional information in the new Figure 10 and added two new Supplementary Figures to help understanding this feature. All the additional information is in Chapter 3f now.

10) In the paragraph beginning at line 222, regarding the range of variability in the two forecast experiments, how do the NAO standard deviations for the individual ensemble members shown in Fig. 4a compare to the observational values? Also, is there any detectable difference in the ensemble spreads between the CTL and EXP forecasts, considering that the latter start with considerable ensemble spread in the land initial conditions?

REPLY: Thank you very much for your comment. We now added an additional Figure in the Supplement showing that the temporal standard deviation of the NAO index from the individual members in the model is very much in line with the ERA20C NAO index deviation. Additionally the new Figure shows the standard deviation among all members (read as member spread) between the 51 CTL members and 21 EXP members. Over the 110 years, the EXP members show higher variability in standard deviation, but the median over all 110 years is virtually the same, see lines 226-228.

11) The legends in Fig. 4 say AFS instead of ASF.

REPLY: Fixed

12) At line 242, please include a reference in order to provide some context about the early twentieth century Arctic warming, such as Polyakov et al. 2003, https://doi.org/10.1175/1520-0442(2003)016%3C2067:VATOAT%3E2.0.CO;2

REPLY: Added the reference

13) The placement of "as depicted in Figure 4" in line 244 suggests that what is being referred to is the depiction of the positive dipole snow pattern in Fig. 3a. If indeed this is referring to Figure 4 then it would be better placed at the end of the sentence.

REPLY: Fixed to Figure 3

14) One or more citations should be added to the sentence in 254-257 (even if cited

previously).

REPLY: Added a reference

15) At line 281, regarding "significantly reduced geopotential heights over the extratropics", according to Fig. 7b these changes are only statistically significant over the Mediterranean region. Please reword accordingly.

REPLY: Reworded

16) Line 301 should state e.g. that anomalies between the two ensemble means in Fig. 7h are "less than" 1 hPa rather than "around" 1 hPa, considering that they never exceed the 0.83 hPa contour level. (Similarly in line 403.)

REPLY: Reworded

17) More attention should be drawn to the different color scales in Figs. 6 vs 7, and 5 vs 8 in order to keep the differences in the magnitudes of the observed and modeled show dipole responses in perspective.

REPLY:We highlight the systematic differences in those two approaches (see reply to your comment 24) now in the discussion section to make the reader more aware of the two very different approaches.

18) It seems noteworthy that the stratospheric evolution in Fig. 8 is similar to, but delayed with respect to that in Fig. 5. This merits at least mentioning, as would any hypotheses the authors might have for the origins of the delay.

REPLY: We now mention this fact at the end of section 3d. Since the atmospheric initial conditions are identical among all the members, it takes some time for the diverging behavior linked to land initialization to manifest.

19) Although the caption to Fig. S2 says "Shading indicates 90% significance level", there is in fact no shading. Does this mean that none of the anomalies are statistically significant?

REPLY: This is correct. None of the anomalies are statistically significant due to large spatial averaging.

20) Is the heading in line 332 intended to be there (and to be labeled "e.")?

REPLY: The heading was intended to be there, but our formatting was faulty. We now formatted the heading accordingly.

21) The term "resulted in" on lines 352 and 354 may imply causality more strongly than intended considering the context of opposite NAO responses to the same sign of snow anomalies. Suggest changing to "preceded".

REPLY: Reworded throughout the manuscript

22) The terminlogy describing the snow anomalies relating to Fig. 10 is inconsistent and somewhat confusing in that the main text in lines 351-352 refers to "high-minus low snow cover anomalies" and the Fig. 10 caption to "positive snow dipole forcing". It would be preferable if consistent terminology describing snow anomalies were used, as addressed also in comments (4) and (5).

REPLY: We fixed the inconsistent wording to "high-dipole" and "low-dipole" throughout the manuscript. We fixed the inconsistent wording to "west-east snow depth gradient" throughout the manuscript.

23) The composite November 1 SST and sea ice concentration differences shown in Fig. S4 which are around 0.1C and 0.02 respectively seem unlikely to have any major impact of DJF NAO. Also, do the authors have confidence that pre-satellite sea ice concentrations during 1901-1978 in ERA20C are sufficiently accurate for such an analysis?

REPLY: Thank you very much for your comment. We agree that sea ice concentration for the period 1901–1978 come with large uncertainties. Nevertheless, we wanted to stay in the ECMWF ecosystem with our analysis. There are other sea ice and SST products available, however they also come with uncertainties. We change the rather absolute wording in the main manuscript to a more relative wording, highlighting that these are the preconditions found in ERA20C, rather than being the preconditions that actually occurred or that are generally found in reality.

24) Do the authors have any hypotheses for why the lagged atmospheric responses to autumn snow dipole index differences are so much weaker in the hindcasts than in ERA-20C? In particular, could this be related to the "signal-to-noise paradox" whereby circulation responses to radiative and surface forcings, as well as the predictability of the NAO, appear to be much weaker in models than is observed as argued by Scaife and Smith 2020, https://doi.org/10.1038/s41612-018-0038-4 ?

REPLY: Thank you very much for your comment. There are several reasons as to why the absolute response is different in magnitude. 1) What you refer to as ERA20C is a linear regression outcome, which units are "per Standard Deviation" and not raw physical units. As such, they are only indirectly comparable to raw units. 2) The regression with ERA20C represents ONE deterministic reality and any results out of one "member" should be stronger than any ensemble mean. 3) Our high and low dipole ensemble means incorporate a wide array of snow states (rather than one deterministic snow state per November), cancelling out possible effects and dampening the overall strength of response. 4) With the regression we sample possible (invisible) covariates with snow. Those covariates might strengthen the NAO response. Not 100% of the regression field we see might be attributable to snow. 5) ERA20C being a reanalysis, we can expect higher variability and better representation of reality than in models and added to that 6) land surface –troposphere–stratosphere interactions in the model are highly likely to be very imperfect. And as such, the model might substantially underestimate the impact of snow depth variability on the wintertime NAO in its model world. 7) As you pointed out, the "signal-to-noise paradox" always impacts predictability studies with dynamic forecast models.

Minor:

lines 141 and 170: a-forementioned -> aforementioned

REPLY: Fixed

line 153 identically -> identical

REPLY: Fixed

line 211: computes -> "represents" or "indicates"

REPLY: Fixed

line 297: mirrors -> mirror

REPLY: Fixed

line 298: increased -> positive

REPLY: Fixed

line 336: pre-ceding -> preceding

REPLY: Fixed

line 379: is -> does ?

REPLY: Fixed

line 383: remove 2nd "the" ?

REPLY: Fixed

line 412: "re-enforced" and "re-inforced" -> reinforced (also line 419)

REPLY: Fixed

line 413: pre-cedes -> precedes

REPLY: Fixed

line 423: mi-  -> mid-

REPLY: Fixed

line 424: remain -> remains

REPLY: Fixed

line 424: suggest "dominant" -> "prominent" (or else simply "remains")

REPLY: Fixed

line 447: "On the opposite" -> "Oppositely" or "On the contrary"

REPLY: Fixed

line 482: could remove "additional"

REPLY: Fixed

line 488: suggest adding "as in this study" after "years"

REPLY: Fixed

line 740: heighst -> heights

REPLY: Fixed

Caption to Fig. S3: "< 1 stand. dev" -> "< -1 stand. dev."

REPLY: Fixed

List of relevant changes:

1) Added additional information and clarity on the non-linear processes concerning the impact of snow depth distribution in Chapter 3f, including new information in Figure 10.

2) Added additional information on the impact of sea ice and tropical precipitation on the DJF NAO in our study.

3) Changed Figures 1,2,3,5,6,9,10 for easier readability.